# Quantitative genetic analysis of respiratory function and related traits in Bulldogs, French Bulldogs and Pugs

Joanna Jadwiga Ilska[1]*, Fern McDonnell[1], Jane Frances Ladlow[2,3]

**1** Health and Breeding Department, Royal Kennel Club, London, United Kingdom, **2** Granta Veterinary Specialists, Linton, United Kingdom, **3** Department of Veterinary Medicine, University of Cambridge, Cambridge, United Kingdom

* joanna.ilska@royalkennelclub.com

## Abstract

Brachycephalic Obstructive Airway Syndrome (BOAS) is a common health issue in brachycephalic breeds like Bulldogs, French Bulldogs, and Pugs, linked to their distinctive skull morphology. Despite its prevalence, the genetic basis of respiratory dysfunction in these breeds remains poorly characterised. To enable selection against BOAS, the Respiratory Function Grading Scheme (RFGS) was established in 2019, where respiratory function of dogs considered for breeding is tested via a standardised exercise test. Here, we analysed RFGS data from over 4,000 dogs, alongside pedigree records, to estimate heritability of respiratory function and assess RFGS participation across the UK Royal Kennel Club registered populations of the three extreme brachycephalic breeds. Moderate heritability estimates for RFGS grade (0.21–0.49), and nostril stenosis (0.31–0.39), with significant genetic correlations between the traits indicate that within-breed selective breeding can improve respiratory health. These findings support the feasibility of breeding programs targeting respiratory function. Implementing such strategies, alongside increased health screening participation, may help mitigate BOAS prevalence and enhance welfare in these popular breeds.

## Introduction

Brachycephalic Obstructive Airway Syndrome (BOAS) is widely regarded as one of the highest priority conformation-related disorders in canine health [1–7]. The condition is causally linked to brachycephalic skull morphology, characterised by a shortened muzzle, wide head and thick neck [8–10]. Clinically, BOAS presents as noisy breathing (stridor and stertor), inspiratory dyspnoea, increased inspiratory effort, regurgitation, heat and exercise intolerance, sleep-disordered breathing, cyanosis, collapse and, in extreme cases, death [8,11]. Although occasionally observed in juvenile dogs (8,12), BOAS is typically diagnosed by three to four years of age [12,13].

**Data availability statement:** All relevant data are within the paper and its Supporting information files.

**Funding:** The author(s) received no specific funding for this work.

**Competing interests:** This study was funded internally by the Royal Kennel Club, where JJI and FMcD are employed.

While owners report that the condition progresses with age [14], several studies have found no significant association between BOAS severity and age in adults [10,11,15–17]. In contrast, a consistent link exists between BOAS and body condition score (BCS), with obese dogs at higher risk of developing the disease, and with greater clinical severity [9–11,16,17].

Although the presenting signs of BOAS were recorded in sources from around 1900 [18], and it was formally described in the1940s [19], its prominence has increased sharply since 2000, coinciding with the surge in global popularity of Bulldogs, French Bulldogs and Pugs – the breeds most frequently affected [9,16,20]. Reported prevalence of BOAS in those breeds varies widely, from as low as 3.5% [21] to over 60% [17,22], depending on diagnostic criteria and sampling approach. Owner-reported estimates likely understate the prevalence, as clinical signs – such as noisy breathing or exercise intolerance – are often normalised as "typical for the breed" [11,23–25]. Conversely, clinic-based data may over-represent affected dogs because healthy brachycephalic dogs are presented less frequently for routine care [26]. Nearly one in six dogs in the UK is currently brachycephalic [27]. This rapid rise in ownership forms an important backdrop to the current welfare debate. Aesthetic appeal remains the principal driver of demand, while health and welfare considerations are often secondary [26,28–30]. Combined with the high prevalence of BOAS, this disconnect between owner perception and clinical reality has been described by some authors as the "brachycephalic crisis" [31].

In response, several interventions have been proposed, including amending breed standards to remove language promoting exaggerated conformation [4,9,23,28,32–34]. In the UK, breed standards for all three breeds were substantially revised in 2009 and iteratively further amended since in an effort to discourage exaggeration (A. Skipper, personal communication). However, the impact of these changes on the overall population remains unclear. Outcrossing programmes have also been proposed to reintroduce more moderate skull shapes [9,33]. The Netherlands attempted to legislate this strategy in 2020 by banning breeding of dogs with a craniofacial ratio (CFR) below 0.3 [6]. Although a recent survey of primarily non-brachycephalic dog owners showed a preference for more moderate appearance [35], previous surveys across the UK, Denmark, and Germany show strong continuity of breed preference among brachycephalic owners [25,26,30,34,36,37]. This likely poses a barrier to population-wide morphological change through outcrossing.

An alternative route for reducing incidence of BOAS involves within-breed selective breeding programmes. A key consideration for such programmes is identifying the phenotype that offers the greatest potential for effective selection. Although CFR captures an important aspect of brachycephalic conformation, it is unlikely to be useful for within-breed selection due to limited phenotypic variation [9,33], and its consistently low predictive value for BOAS [10,17,38,39], as well as low reproducibility [17]. Despite low average CFR values, ranging from 0.11 (SD 0.04) in Pug to 0.16 (SD 0.04) in French Bulldog [17], not all dogs in those breeds are clinically affected, indicating additional risk factors. To date, no estimates of genetic correlations between CFR and BOAS have been published.

These limitations highlight the need for phenotype assessments that capture functional consequences of conformation rather than conformation itself. Objective assessments using whole-body plethysmography (WBBP) accurately characterise respiratory function [11], but the method's cost and technical requirements preclude population-level use. An efficient pre-breeding screening protocol must be accurate, minimally invasive, quick to administer, inexpensive and widely accessible. Consequently, several exercise-based functional grading systems have been developed that quantify respiratory distress through pre- and post-exercise auscultation [11,15,16,38–42].

The Royal Kennel Club (RKC), in collaboration with the University of Cambridge, introduced the Respiratory Function Grading Scheme (RFGS) in 2019, following formal validation of an exercise and auscultation method against WBBP [11,16,42]. Validation studies demonstrated strong correlations between post-exercise clinical grades and WBBP-derived indices [16]. The exercise challenge used in RFGS reveals cases that may be underestimated at rest, highlighting the advantage of dynamic assessment [11]. The RFGS assigns functional grades from 0–3, with grades 0–1 considered clinically unaffected and grades 2–3 indicating BOAS of increasing severity. High inter-assessor repeatability and the clear differentiation between unaffected and affected dogs further support the method's robustness [42]. The RFGS therefore provides a practical, validated system suitable for large-scale screening and for investigating phenotypic severity and heritability of BOAS traits. Full RFGS test procedures and breeding guidance are provided in S1 File.

Current breeding recommendations under RFGS were based on pragmatic selection against a highly prevalent trait presumed to be heritable, despite limited information on its genetic architecture. Several genes have been linked to brachycephalic skull morphology, e.g., SMOC2 – which accounts for 36% of phenotypic variation in facial length [43], BMP3 [44], or DVL2 [45,46]. However, these variants are nearly fixed in extreme brachycephalic breeds [44,45], suggesting additional genetic or morphologic contributors to BOAS. A genetic basis is nonetheless strongly supported: two currently available estimates of heritability for BOAS range from 0.4 with a considerable standard error of 0.16 using mid-parent-offspring regression [47] to 0.6 (no standard error of the estimate provided) based on preliminary genome-wide association study of approximately 200 dogs per breed [48].

The aims of this study were to estimate the heritability of respiratory function grades in Bulldogs, French Bulldogs and Pugs, and to evaluate the potential for within-breed selection against BOAS in UK populations registered with the RKC. To achieve this, we analysed two complementary datasets: routine pre-breeding RFGS records from the largest cohort studied to date, and a secondary research dataset with additional phenotypes. Finally, we quantified participation in the RFGS across subpopulations, distinguishing between show-bred dogs and those bred outside the breed standard (based on coat colour), to evaluate the potential for improving upper respiratory health in these three popular brachycephalic breeds.

## Materials and methods

### Data

Data used in the analyses included 1) grades collected through the RFGS on Bulldog, French Bulldog and Pug; 2) additional research data collected on some of the RFGS tested dogs in the three breeds, 3) complete pedigree records for the three breeds as available in the electronic database of the RKC. The data is available in S3–S8 Tables.

The RFGS dataset included a total of 4301 grades, of which 1474, 1917 and 900 were recorded on Bulldog, French Bulldog and Pug respectively. The dataset included identification of the dog (RKC registration number, RKC registered name, microchip and RKC internal dog reference number), dog's date of birth, RFGS test date, assessor and grade recorded on a scale from 0 to 3 as described previously [42]. The assessor identifier was missing for 583 grades, and these were assigned unique dummy variables. For dogs with multiple grades available, the number of days between first and last test was calculated, as well as grade progression classified as improved, same, or worsened. Although RFGS was officially launched in 2019, there were 73 grades collected during Scheme development between 2015 and 2018.

RFGS grades were converted to binary BOAS status following [17] for summary reporting, with grades 2 and 3 treated as clinically affected (BOAS+). Date of birth (DOB) and date of test (DOT) were converted to ordinal format then scaled

and centred, and age at testing was calculated in days (scaled and centred), with age in completed years used for descriptive summaries. Year of birth and year of test were extracted from DOB and DOT, and centred.

For genetic analyses, the ordinal RFGS grades were transformed to presumed underlying liability scale using the threshold model [49] as shown in Eq 1.

$$y_j = \frac{\phi(z_{j-1}) - \phi(z_j)}{\Phi(z_j) - \Phi(z_{j-1})}, \ z_j = \Phi^{-1}(\textstyle\sum_{i=1}^{j} p_i)$$

(1)

Where $y_j$ is the expected value of underlying liability distribution for individuals with grade $j$, $p_j$ is the observed proportion of grade $j$, $\phi()$ and $\Phi()$ are the standard normal probability density and cumulative distribution functions respectively.

The additional research dataset included some of the RFGS dogs, with variable levels of recording per individual and included: 2248 body weight measurements (BW, kg), 3084 body condition scores (BCS) on a 9-point scale with increasing scores reflecting increasing body fat content [50], and 3133 nostril stenosis (STEN) graded on a 4-point scale of open, mild, moderate and severe stenosis, as described previously [17]. A binary obesity variable was generated, classifying dogs with BCS ≥ 7 as obese. STEN grades were converted to underlying liability scale as previously shown for RFGS grade (Eq 1) for the genetic analyses, and to binary stenotic nares status for summary reporting, with dog treated as affected, i.e., presenting with stenotic nares, when the STEN grade was moderate or severe [17].

Pedigree included dog's sex, date of birth, coat colour, studbook number (SBN, awarded to dogs successful in conformation shows) if available, and registration type, as well as identification of dog's sire and dam. To identify dogs bred primarily for conformation shows, show breeding (SB) was calculated as twice the number of parents with SBN, plus the number of grand-parents with SBN, and so ranged between 0, where none of ancestors in last two generations had a SBN, and 8, where all ancestors had an SBN. The pedigree was checked and corrected for inconsistencies in parentage using custom Fortran 90 program. Inbreeding coefficients for all dogs in the pedigree were calculated using a Fortran 90 program following the algorithm of [51].

The coat colour was grouped into colours recognised in the breed standard for the breed (BS), and other colours labelled as non-breed standard (NBS). The list of coat colours and their assignment to the BS and NBS categories is shown in S1 Table.

To examine possible sample bias, statistics pertaining to show breeding and coat colours were compared between RFGS tested dogs and their contemporaries, selected from pedigree as RKC-registered dogs resident in the UK and born in the same period as tested dogs – with the cut-off date being the date of birth of the oldest tested dog in each breed. Similarly, percentage of RFGS tested dogs and percentages of litters born to tested parents were calculated based on the data for RKC-registered and UK-resident dogs.

## Summary statistics

As the majority of the variables recorded were ordinal and showed substantial deviation from Gaussian assumptions, a range of statistical tests were used to ensure a robust inference. Between breed differences in RFGS grade, BCS, STEN grade, age in years, and the number of days between first and last record for dogs with multiple results, were assessed using Kruskal-Wallis rank sum test. Pairwise differences between breeds in RFGS grade and age in years were tested using Wilcoxon rank sum test, with Bonferroni correction for multiple testing. Wilcoxon rank sum test with continuity correction was used to assess if there was evidence of selection based on RFGS based on the difference in first available RFGS grade distribution between breeding dogs and dogs without registered offspring. To assess whether RFGS testing influenced sire selection, a two-sample t-test with unequal variances was used to compare the number of litters produced by tested versus untested sires. Between breeds sex distribution, BOAS prevalence, percentage of NBS dogs, count of dogs in show breeding categories and disease progression between first and last result were tested using Pearson

Chi-squared test, followed by Chi-squared test with Bonferroni adjustment for multiple testing for pairwise breed differences. Within breed differences in mean body weight between males and females were assessed using t-test. Logistic regression models were used to assess odds of being tested given breed and show breeding, and to assess odds of being BOAS+ given coat colour group, obesity and STEN in each breed. Models were fitted with a binomial error distribution and logit link. Odds ratios (OR) with 95% confidence intervals (CI) were obtained by exponentiating regression coefficients. Where relevant, pairwise contrasts were estimated using marginal means with p-value adjustment for multiple testing.

## Population structure

Population structure of each breed was examined through visual appraisal of the first two principal components (PC1 and PC2) calculated using randomised singular value decomposition in randPedPCA [52]. Scatter plots of PC1 and PC2 were generated with individuals coloured according to key factors relevant to this study -year of birth, coat colour group, show breeding and RFGS tested status. This allowed us to assess whether these variables corresponded to clustering patterns within each breed.

## Animal models

Preliminary analyses involved heritability estimation using parent-offspring regression. Details of the method, models, and results are in S2 File. Final estimates were obtained from a mixed linear animal model incorporating all known pedigree relationships. Mixed linear univariate models were fitted to the RFGS data for each breed using ASRemlR [53] to assess the presence and magnitude of genetic variance in RFGS grades in Bulldog, French Bulldog and Pug, and to calculate estimated breeding values (EBVs). Separately, univariate mixed linear models were fitted to BW, BCS and STEN in each breed, and a log-likelihood ratio test as implemented in ASRemlR (LRT) was used to identify which of these traits had a significant genetic variance. A multivariate mixed linear model was fitted to RFGS grade and additional traits for which significant genetic variance was detected to calculate correlations between traits.

The univariate models fitted to the transformed RFGS grade, BW, BCS and the transformed STEN were:

$$y = X\beta + Z_1u_1 + Z_2u_2 + Z_3u_3 + f(x) + e \qquad (2)$$

where $y$ is a vector of phenotypes, $X$ is the design matrix for fixed effects and covariates described below: $Z_1$, $Z_2$ and $Z_3$ are incidence matrices relating observations to the additive animal ($u_1$), permanent environment ($u_2$) and assessor ($u_3$) effects, all assumed random, $f(x)$ represents a smooth non-linear function of age, modelled using a natural cubic spline, and $e$ is the vector of residuals. Random terms were assumed to be multi-variate normally distributed, with parameters $u_1 \sim N(0, A\sigma_a^2)$, $u_2 \sim N(0, I\sigma_{pe}^2)$, $u_3 \sim N(0, I\sigma_{as}^2)$, and $e \sim N(0, I\sigma_e^2)$, where $A$ is the numerator relationship matrix, $I$ is an identity matrix, and $\sigma_a^2$, $\sigma_{pe}^2$, $\sigma_{as}^2$ and $\sigma_e^2$ are variances corresponding to additive, permanent environment, assessor and residual terms respectively. The numerator relationship matrix was calculated from the pedigree restricted to tested dogs and their ancestors, using trimPed function from *pedigree* package [54]. Each of the random terms was tested using LRT, and with exception of splines, only terms significant for the trait in each breed were used in further analyses.

All models included sex fitted as a fixed effect, and a linear effect of age (in days) fitted as a covariate – in addition to the previously described random splines on age. Other terms: coat colour group, SB, year of birth and testing, ordinal date of test, coefficient of inbreeding (*f*) and protocol (for Bulldog and French Bulldog only), were tested through backwards elimination with Bayesian Information Criterion (BIC) used as a determinant. The tested terms were removed from the model if their removal resulted in BIC improvement of 2 or more [55]. In the analyses of additional research data, traits which did not have a significant genetic variance and obesity were added to the list of effects to be tested through backwards elimination.

Once the fixed effects and random terms were finalised for the univariate analysis of transformed RFGS grades, narrow sense heritability was calculated as proportion of the phenotypic variance explained by $\sigma_a^2$. Phenotypic variance was calculated as a sum of $\sigma_a^2$ and $\sigma_e^2$, with addition of $\sigma_{pe}^2$ in models where this component was significant. In those models, repeatability was also calculated, as a proportion of the phenotypic variance explained by the sum of $\sigma_a^2$ and $\sigma_{pe}^2$. The assessor term and spline on age were treated as a nuisance factor and were not included in the calculated phenotypic variance.

The final model was re-run using the entire pedigree to calculate EBVs for all individuals. Prediction error variances (PEV) were computed as the squared standard errors of EBVs, and the accuracies of the EBVs were calculated as:

$$r = \sqrt{1 - \frac{PEV}{(1+f)\sigma_a^2}}$$

(3)

Where $f$ is the inbreeding coefficient. Multivariate mixed models were fitted within breed to estimate the correlations between transformed RFGS grade and research data traits with significant genetic variance. Random terms were assumed to be normally distributed, with parameters $u_1 \sim N(0, A \otimes G)$, $u_2 \sim N(0, I \otimes C)$, and $e \sim N(0, I \otimes R)$, where $G$ is the additive genetic (co)variance matrix across the traits, and $C$ and $R$ are assessor and residual (co)variance matrices respectively. No permanent environment covariances between traits were estimated. Fixed and random effects fitted to each trait were as stated for the univariate analyses. The significance of genetic and residual correlations between pairs of traits in each breed were tested using log-likelihood ratio test as implemented in ASRemlR [53].

Phenotypic correlations between traits $i$ and $j$ were calculated as:

$$r_{P_{ij}} = \frac{\sigma_{a_{ij}} + \sigma_{e_{ij}}}{\sqrt{(\sigma_{a_i}^2 + \sigma_{pe_i}^2 + \sigma_{e_i}^2) \times (\sigma_{a_j}^2 + \sigma_{pe_j}^2 + \sigma_{e_j}^2)}}$$

(4)

Where $\sigma_{a_{ij}}$ and $\sigma_{e_{ij}}$ were additive genetic and residual covariances, and $\sigma_{pe_i}^2$ was a permanent environment variance fitted only in traits where it was deemed significant in univariate analyses. The estimates and standard errors of the phenotypic correlations were obtained using *vpredict* function in ASRemlR [53]. Phenotypic correlations between traits were tested for significance using the classical Fisher correlation test, implemented via the exact t-distribution for Pearson's r, as $t = r\sqrt{\frac{(n-2)}{(1-r^2)}}$, where $r$ denotes the correlation estimate, and $n$ is the number of observations where both traits $i$ and $j$ were recorded.

## Results

### Summary statistics for RFGS data

The prevalence of clinical BOAS was moderately low and differed significantly between breeds. French Bulldog had the lowest prevalence at 15.6%, compared with 18.9% in Bulldog ($\chi^2$=6.1, p_adj=0.04) and 19.8% in Pug ($\chi^2$=7.1, p_adj=0.02), with no significant difference between Bulldog and Pug ($\chi^2$=0.2, p_adj=1). This corresponded to 26% (OR=1.26, CI: 1.05 to 1.51, p=0.012) and 33% (OR=1.33, CI: 1.08 to 1.63, p=0.012) higher odds of clinical disease in Bulldog and Pug respectively, compared with French Bulldog. RFGS grade distribution (Fig 1) also differed between breeds ($\chi^2$=79.2, p<0.001), driven by French Bulldog, whose distribution differed significantly from both Bulldog and Pug (p_adj<0.0001), while Bulldog and Pug did not differ (p_adj=0.15). The summary statistics pertaining to the RFGS grades and dogs participating in the Scheme are presented in Table 1.

There was a significant difference in distribution of age measured in years between breeds ($\chi^2$=86.9, p<0.0001). Pug dogs were significantly older than both Bulldog and French Bulldog (p_adj<0.0001), and the difference between the latter two breeds was less pronounced (p=0.045). There was no significant difference in sex distribution ($\chi^2$=4.2, p=0.123), with females over-represented in all three breeds.

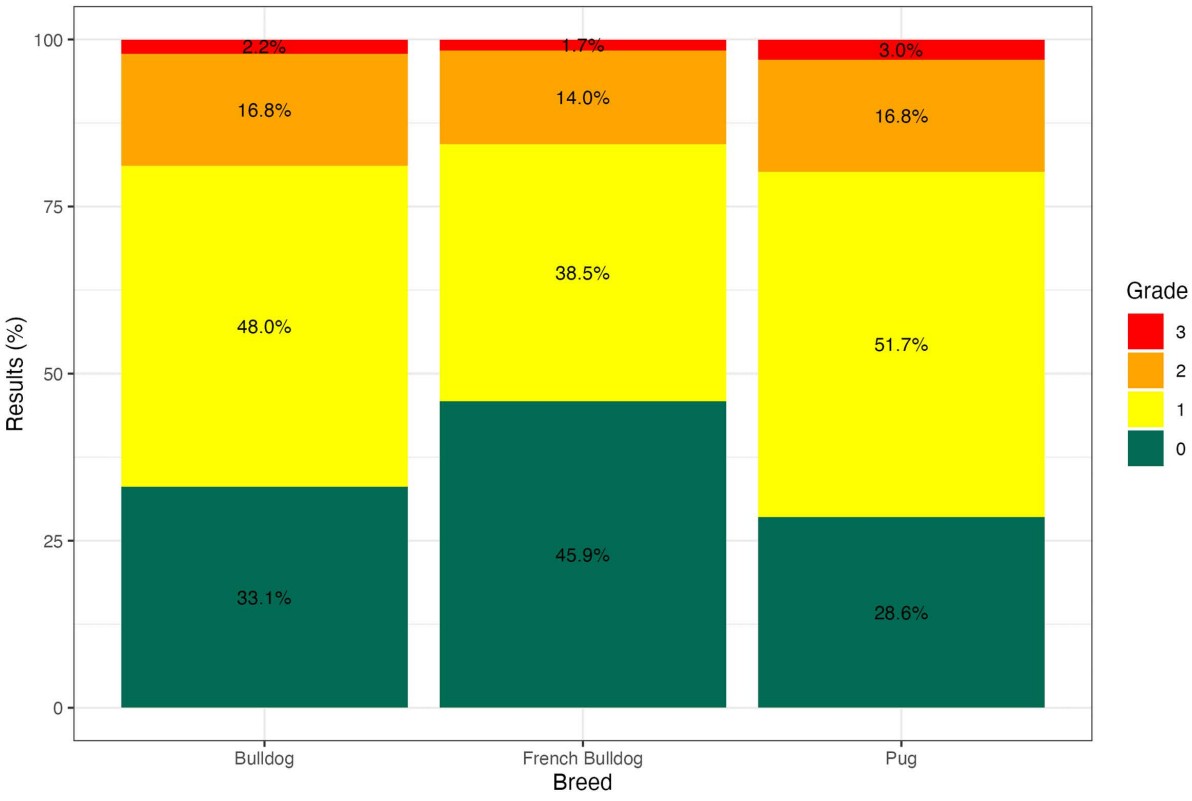

**Fig 1. Distribution of RFGS grades between the breeds.**

Only a small percentage of dogs in each breed were tested under the RFGS. As shown in Fig 2A, the highest testing rate was found for Pug dogs born in 2023 (3.1%). Rates in French Bulldog and Bulldog remained low and stable since scheme inception in 2018, at 0.4–0.9%, and 1.3–1.7% respectively. In contrast, the percentage of litters sired by RFGS-tested dogs (Fig 2B) increased steadily, reaching 33.8% of Pug litters born in the first half of 2025, followed by 26.9% in Bulldog and 9.5% in French Bulldog. Litters with both parents tested remained rarer at 17.5%, 13.2% and 4.3% for Pug, Bulldog and French Bulldog respectively.

There was modest evidence of phenotypic selection based on RFGS testing status and grade. In all three breeds, approximately half of the tested dogs were bred from (52.1%, 52.6% and 55.5% in Bulldog, French Bulldog and Pug respectively), and the number of litters produced by tested sires was significantly higher than that of untested sires (Table 2). In Bulldog and French Bulldog, breeding dogs had significantly lower mean first available RFGS grade than non-breeding dogs, although differences were small (Table 2). In Pug, there was no statistically significant difference in the distribution of RFGS grades between breeding and non-breeding dogs.

Across the RFGS data, the majority (n = 3274) of the dogs were tested only once. However, nearly a quarter (23.6%) of the RFGS grades were recorded on dogs with multiple records, including 192 males contributing 423 records, and 271 females contributing 589 records. Most (n = 385) dogs in this group were tested twice. The highest number of results recorded on a single dog was 7, with all results showing the same RFGS grade. Comparing the first and last available record, RFGS grades remained the same for the majority of the dogs (n = 274), followed by 101 dogs for which last grade was better than first, and 88 dogs for which grade worsened over time. The progression of RFGS grades differed

**Table 1. Summary statistics for the RFGS data.**

| | Bulldog | French Bulldog | Pug |
|---|---|---|---|
| *Number of results* | 1,474 | 1,917 | 900 |
| Male | 35.1% | 38.4% | 37.7% |
| Female | 64.9% | 61.6% | 62.3% |
| Age at testing in completed years – Median (Range, and IQR) | 1.5 (1–10, IQR = 1) | 1 (1–12, IQR = 1) | 2 (1–12, IQR = 2) |
| *Number of unique dogs* | 1,303 | 1,691 | 743 |
| *Male* | 33.6% | 38.0% | 37.6% |
| *Female* | 66.4% | 62.0% | 62.4% |
| Coat colour; UKBreed (% of UKBreed tested) | | | |
| Breed Standard | 81,354 (1.5%) | 150,454 (0.8%) | 105,757 (0.7%) |
| Non-Breed Standard | 39,433 (0.2%) | 192,462 (0.3%) | 11,523 (0.3%) |
| Unknown | 15 (6.7%) | 87 (0.0%) | 1 (0.0%) |
| Show Breeding; UKBreed (% of UKBreed tested) | | | |
| 0 | 72,027 (0.3%) | 324,720 (0.3%) | 84,441 (0.2%) |
| 1 | 19,857 (0.9%) | 8,298 (1.5%) | 15,803 (0.4%) |
| 2 | 10,521 (1.8%) | 4,643 (3.1%) | 5,730 (1.4%) |
| 3 | 8,754 (2.4%) | 2,648 (4.2%) | 4,293 (1.8%) |
| 4 | 6,229 (3.9%) | 1,491 (7.3%) | 3,793 (2.8%) |
| 5 | 2,370 (5.7%) | 523 (9.8%) | 1,755 (4,7%) |
| 6 | 686 (9.6%) | 422 (11.4%) | 858 (7.9%) |
| 7 | 288 (10.1%) | 194 (19.6%) | 461 (14.5%) |
| 8 | 70 (10.0%) | 64 (18.8%) | 150 (19.3%) |

IQR – Inter-quartile range; UKBreed - number of UK-resident dogs in particular category, born in the same period as tested dogs; % of UKBreed tested – indicates the percentage of UKBreed dogs which were RFGS tested.

significantly between breeds (p < 0.001), however, so did the number of days between first and last test (p < 0.001), as shown in Table 3.

## Summary statistics for the research data

The summary statistics for BW, BCS and STEN are shown in Table 4. Distribution of BW was approximately normal in all three breeds, as shown in Fig 3A, and in all three breeds males were significantly heavier than females (p < 0.0001). The distribution of BCS (Fig 3B) and STEN differed significantly between breeds (p < 0.0001). Pug had the highest percentage of obese dogs at 16.8%, and French Bulldog had the lowest percentage of dogs with open nostrils at 6.2%.

 Across all three breeds, obese dogs had 3.8x (CI: 2.87–5.09, p < 0.0001) higher odds of being clinically affected by BOAS, with the within breed odds of 3.7x (CI: 2.39–5.82), 2.9x (CI: 1.46–5.76) and 4.5x (CI: 2.75–7.44) for Bulldog, French Bulldog and Pug respectively. Similarly, dogs with stenotic nares had 2.6x (CI: 2.16–3.13, p < 0.0001) higher odds of being clinically affected by BOAS, with the within breed odds of 2.7x (CI: 1.93–3.69), 4.0x (CI: 2.91–5.57) and 2.7x (CI: 1.75–4.16) for Bulldog, French Bulldog and Pug respectively.

## Population structure

The contemporary populations of all three breeds were dominated by dogs with no ancestors winning conformation shows (SB = 0), although the distribution of SB categories differed substantially between breeds ($\chi^2$ = 91,644, p < 0.0001). The

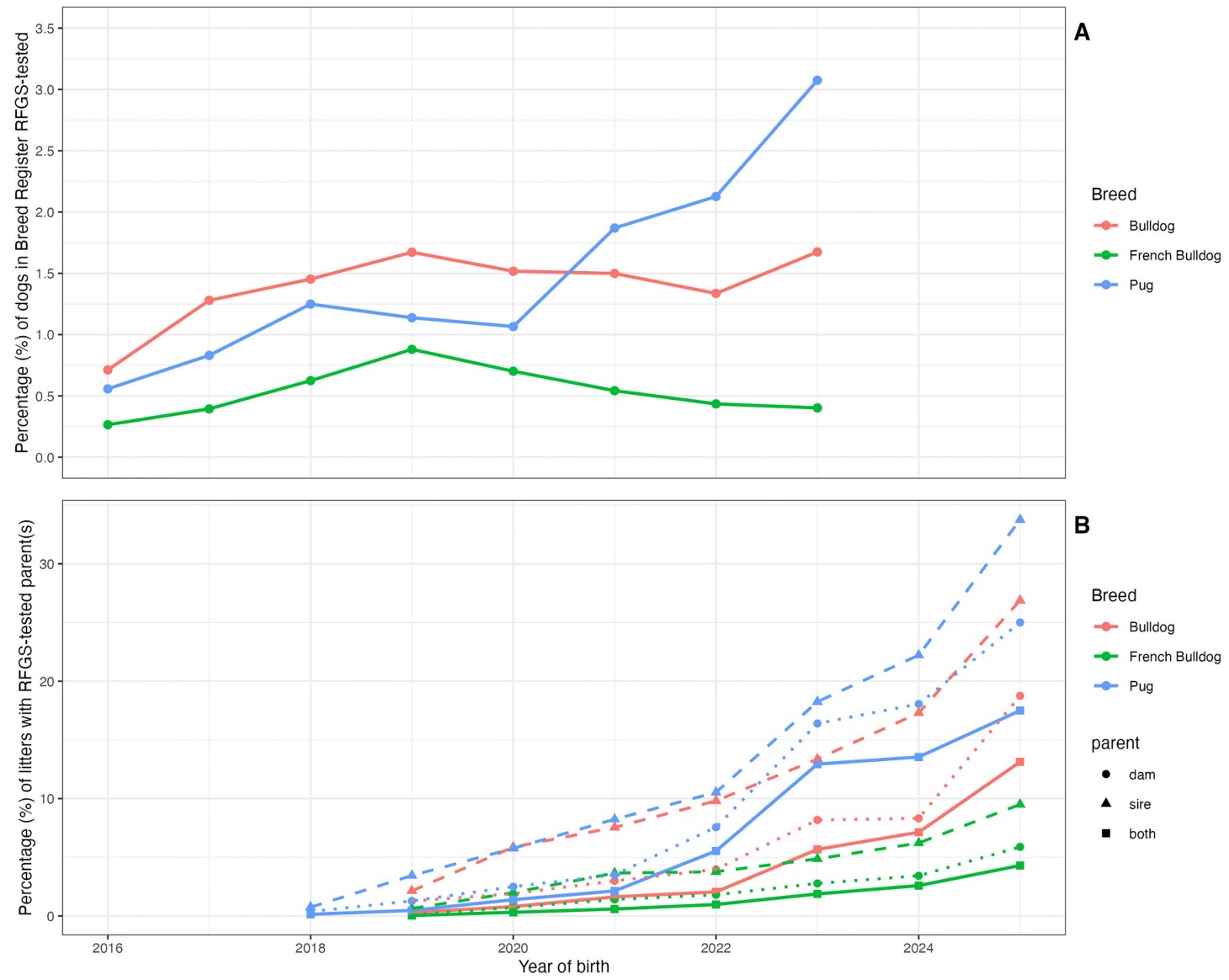

**Fig 2. Percentages of the population tested using RFGS by year of birth.** A – percentage of UK-resident dogs RFGS tested. B – percentage of UK-born litters with either parent, or both parents RFGS tested.

highest percentage of dogs with SB = 0 was observed in French Bulldog (94.7%), followed by Pug (72.0%) and Bulldog (59.6%).

Dogs of non-breed standard colours (NBS) were common within the contemporaries of RFGS-tested dogs, but their percentage varied markedly between breeds ($\chi^2$=82,317, p_adj < 0.0001). The percentages of NBS dogs ranged from 9.8% in Pug, through 32.6% in Bulldog and 56.1% in French Bulldog.

Population structure analyses indicated emerging genetic divergence between coat colour groups within each breed. PCA plots (Fig 4) showed that, within each breed, dogs born since 2016 formed a single contemporary cluster; however, BS and NBS dogs were positioned toward opposite extremes of their respective breed clusters. PCA plots stratified by

**Table 2. Evidence of selection based on RFGS testing status and grade. 1) Mean number of litters per sire, with/out RFGS result, 2) Mean first available RFGS grade for dogs used in breeding and non-breeding dogs.**

| | | Bulldog | French Bulldog | Pug |
|---|---|---|---|---|
| 1. Mean number of litters per sire | Untested sire (SD) | 4.99 (9.65) | 4.82 (10.09) | 5.18 (8.26) |
| | Tested sire (SD) | 8.88 (12.33) | 7.94 (12.92) | 7.40 (9.40) |
| | p-value of difference | <0.001 | <0.001 | 0.007 |
| 2. Mean first available RFGS grade | Breeding dogs (SD) | 0.82 (0.71) | 0.60 (0.66) | 0.94 (0.70) |
| | Non-breeding dogs (SD) | 1.00 (0.82) | 0.89 (0.86) | 1.03 (0.79) |
| | p-value of difference | <0.001 | <0.001 | Not significant |

**Table 3. Number of dogs whose last RFGS grade improved, remained the same or worsened as compared to the first available grade, across the three breeds.**

| Breed | Improved | Same | Worsened | Mean days between tests (±SD) |
|---|---|---|---|---|
| Bulldog | 36 | 92 | 27 | 697.18 (441.47) |
| French Bulldog | 24 | 123 | 31 | 793.75 (425.73) |
| Pug | 41 | 59 | 30 | 933.40 (531.73) |

**Table 4. Summary statistics for additional research data.**

| | Bulldog | French Bulldog | Pug |
|---|---|---|---|
| *Weight (kg) (n)* | 769 | 1,046 | 433 |
| Mean (±SD) | 24.3 (3.61) | 12.0 (2.13) | 8.4 (1.39) |
| Mean (±SD) Dog | 26.8 (3.38) | 13.1 (2.04) | 9.3 (1.35) |
| Mean (±SD) Bitch | 23.0 (2.96) | 11.3 (1.87) | 7.9 (1.16) |
| *BCS (1–9) (n)* | 1,061 | 1,487 | 536 |
| Median (Range, IQR) | 5 (2–8, IQR = 1) | 5 (2–8, IQR = 0) | 6 (4–9, IQR = 1) |
| Obese | 8.5% | 2.6% | 16.8% |
| *Nostril Stenosis* | 1,077 | 1,507 | 549 |
| Open | 23.3% | 6.2% | 14.8% |
| Mild | 54.7% | 42.6% | 45.0% |
| Moderate | 20.7% | 45.2% | 38.1% |
| Severe | 1.3% | 6.0% | 2.2% |

year of birth (S1–S3 Figs) showed that the separation between coat colour groups became more apparent in recent cohorts, coinciding with the expansion in popularity of these breeds. While some temporal dispersion over time is expected due to pedigree including dogs with unknown parentage treated as founders [52], the consistent positioning of coat colour groups toward opposite extremes suggests emerging population substructure. RFGS testing patterns overlapped strongly with show breeding in all breeds.

RFGS participation was strongly associated with show-breeding background, with each step increase in SB raising the odds of being tested by 71% (CI: 66.7 to 75.6) in Bulldog, 97.4% (CI: 92.7 to 102.1) in French Bulldog and 85.0% (CI: 80.0 to 90.0%) in Pug.

RFGS participation was also strongly associated with BS coat colour. In all three breeds, dogs of BS colour were more commonly tested than NBS dogs, with odds ratio of 9.2x (CI: 7.2 to 11.8, p_adj < 0.001) in Bulldog, 3.0x (CI: 2.7 to 3.3, p_adj < 0.0001) in French Bulldog and 2.3x (CI: 1.6 to 3.2, p_adj < 0.001) in Pug. Although the sample sizes of NBS dogs

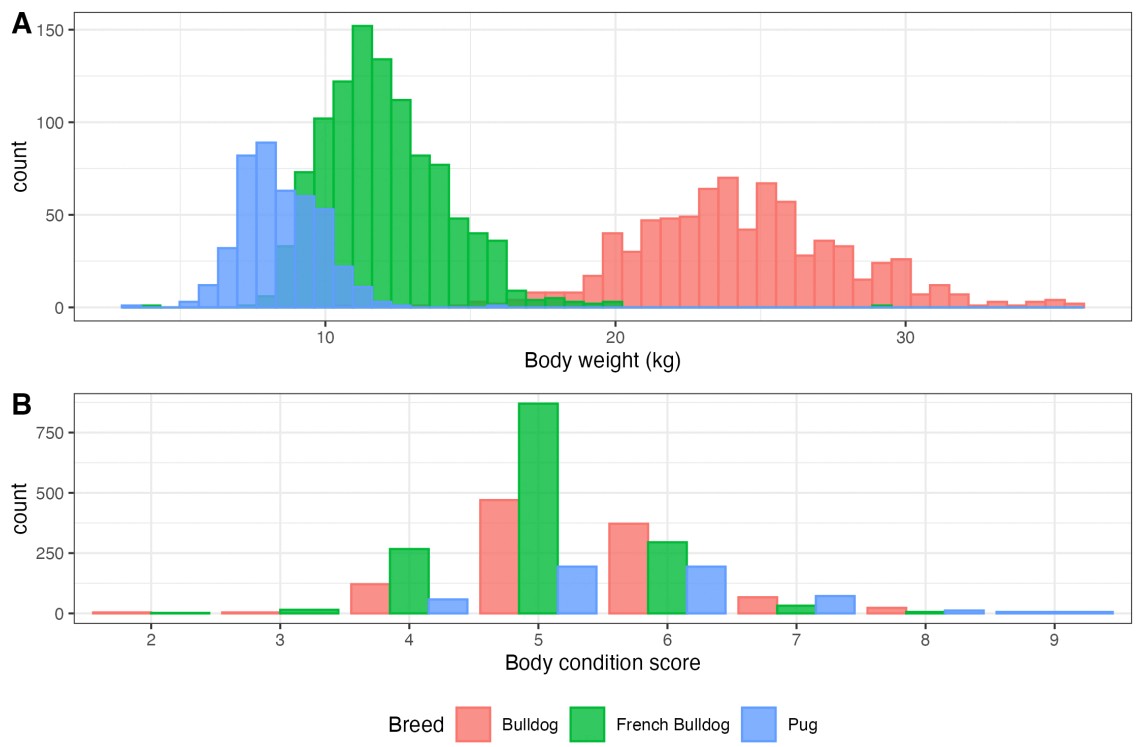

**Fig 3. Distribution of body weight (A) and body condition score (B) across the three breeds.**

were generally small, the odds of being BOAS+ were higher for NBS dogs, at 2.9x (CI: 1.72 to 4.70) in Bulldog, 1.9x (CI: 1.45 to 2.42) in French Bulldog and 3.6x (CI: 1.81 to 7.23) in Pug.

## Genetic parameters of RFGS grade

All three breeds showed significant genetic variance in RFGS grade, with moderate heritability estimates and high repeatability where estimable. Heritability ranged from 0.21 (SE 0.06) in Bulldog to 0.45 (SE 0.06) in French Bulldog, with Pug at 0.28 (0.07). Repeatability was high in Bulldog (0.49, SE 0.07) and French Bulldog (0.61, SE 0.05), while in Pug the permanent environment effect was not significant and thus no repeatability estimate was calculated. Fixed effects structures were the same for Bulldog and Pug (Table 5). Age had a significant but small linear effect only in Pug (0.13 ± 0.03 units per SD increase in age, equivalent to 0.07 per year). Sex was significant in Bulldog and French Bulldog, with females showing lower RFGS grades, whereas in Pug the sex difference was nonsignificant but slightly higher in females. Non-linear age splines were not significant in any breed but were retained for consistency.

EBV accuracy for RFGS grade was generally low across breeds. Distribution of the accuracies for dogs born since 2016 is shown on Fig 5, with median accuracies of 0.24, 0.26 and 0.22 for Bulldog, French Bulldog and Pug. Only a small percentage of contemporary dogs exceeded the accuracy expected from selection on phenotype alone. The dissemination of information from phenotyped dogs through the pedigree differed across breeds: Bulldog showed the widest spread of information, whereas in French Bulldog the percentage of dogs with highly accurate EBVs mirrored the percentage of phenotyped dogs, indicating poor propagation of phenotypic information. Accuracies differed between BS and NBS dogs in Bulldog and Pug, with higher median accuracies in BS dogs; this contrast was most pronounced in Bulldog (0.33 vs

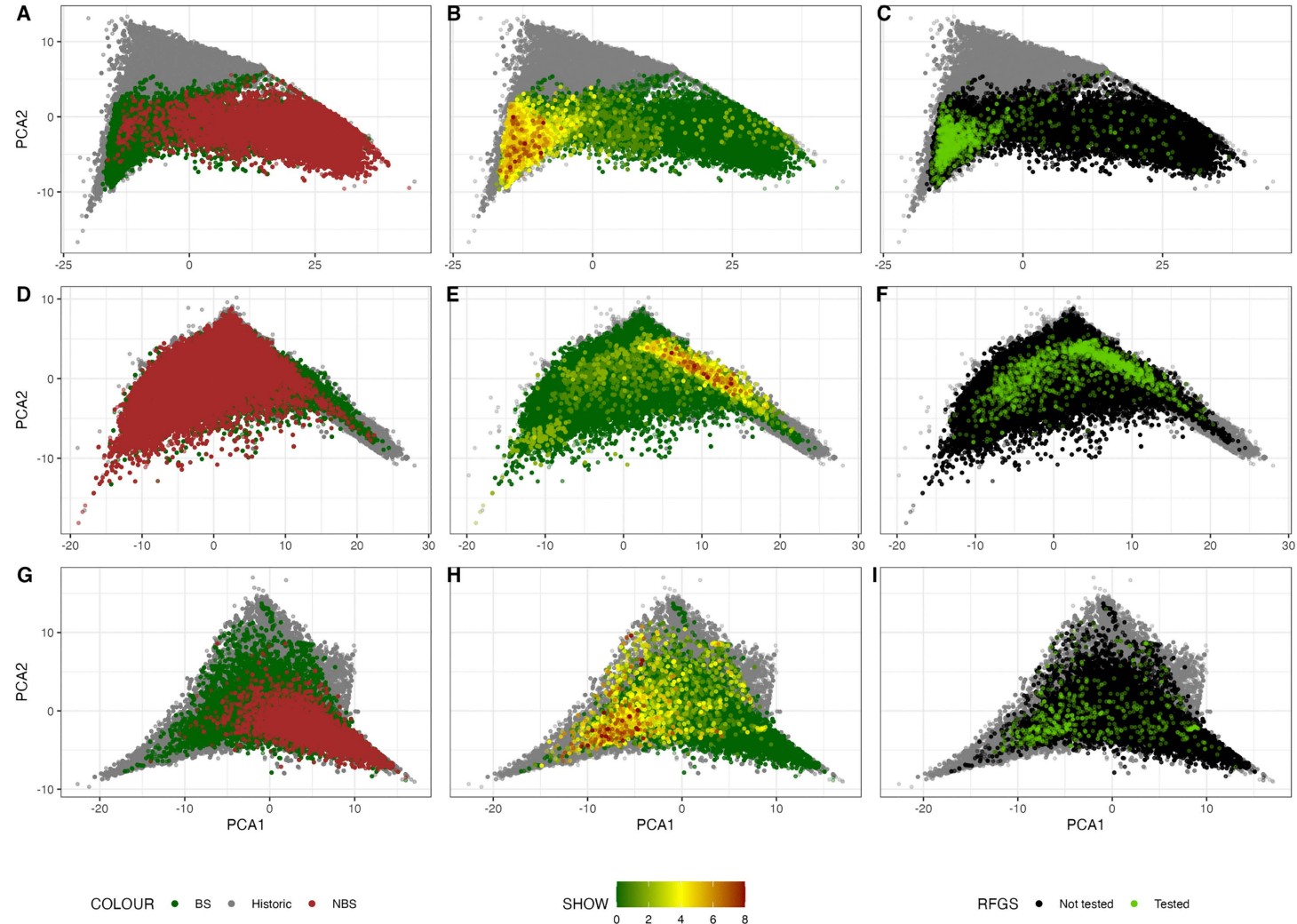

**Fig 4. Principal component analysis (PCA) plot of population structure in Bulldog (A – C), French Bulldog (D – F) and Pug (G – I).** First column illustrates population structure in context of coat colour, second column in context of show breeding and third column in context of RFGS testing status. Dogs marked in grey were defined as historic dogs born before 2016.

0.19), compared to a smaller difference in Pug (0.22 vs 0.19). No difference in median EBV accuracy between colour groups was observed in French Bulldog.

### Correlations between RFGS grade and conformation traits

Multivariate models identified significant genetic variance in STEN grade and BW across all three breeds, and in BCS in French Bulldog, with notable genetic correlations between traits. The strongest significant genetic correlation was observed between STEN and RFGS grades in French Bulldog (Table 6). Assessor effects were positively correlated for RFGS and STEN grades in both Bulldog (0.66, SE 0.19, p = 0.007) and Pug (0.68, SE 0.20, p = 0.015), and in Pug the assessor effect was also shared between BW and STEN grade (0.60, SE 0.23, p = 0.033). Residual correlations were positive and significant for most trait pairs in French Bulldog and Bulldog, except correlation between BW and STEN grade

**Table 5. Fixed effects used in the final models used to derive estimates of RFGS grade heritability in the three breeds. P-values <0.05 marked in bold.**

| Effect | Bulldog Estimate (SE) | p-value[a] | French Bulldog Estimate (SE) | p-value[a] | Pug Estimate (SE) | p-value[a] |
|---|---|---|---|---|---|---|
| Intercept | 0.21 (0.12) | 0.340 | 0.31 (0.08) | 0.193 | 0.08 (0.10) | 0.086 |
| Sex (male as reference) | **−0.30 (0.05)** | **<0.001** | **−0.33 (0.04)** | **<0.001** | 0.08 (0.06) | 0.194 |
| Age in days (scaled and centred) | 0.02 (0.03) | 0.441 | 0.05 (0.03) | 0.128 | **0.13 (0.03)** | **<0.001** |
| Colour group (BS as reference) | 0.26 (0.15) | 0.095 | | | **0.35 (0.17)** | **0.038** |
| f | −0.52 (0.55) | 0.346 | 0.14 (0.59) | 0.817 | −0.54 (0.69) | 0.430 |
| Year of birth (centred) | | | 0.13 (0.07) | 0.071 | | |
| Ordinal date of birth (scaled and centred) | | | −0.23 (0.15) | 0.113 | | |

Intercept represents the expected response for a male of a breed standard colour, with coefficient of inbreeding (f) of 0, of mean age, mean year and date of birth.

[a] P-values were calculated using the Wald test.

(non-significant in both breeds) and RFGS grade and BW, which reached significance only at p < 0.1 in French Bulldog. Full results for residual and assessor correlations are presented in S2 Table.

## Discussion

This study provides the most comprehensive genetic analysis of BOAS-related traits in UK RKC registered populations of the three principal brachycephalic breeds to date. We report prevalence estimates of clinically expressed BOAS that differ from what has been previously documented and demonstrate that within-breed selection has the potential to achieve meaningful genetic improvement. Moderate heritability estimates for RFGS grade, together with additional estimates for stenotic nares and body weight, expand our current understanding of the genetic architecture underlying BOAS and its associated conformational traits. The study also offers practical recommendations for the continued development of the RFGS scheme and provides demographic insights showing population substructure. Further, our data indicate that a large percentage of dogs bred in the UK do not conform to their breed standards, and that participation in health testing, although still limited, is higher among dogs linked to conformation-show lines.

The estimated prevalence of BOAS in our study (16–20%) is substantially lower than previously reported [11,16,17,22,38,40,47]. Because participation in RFGS testing is voluntary, the true prevalence is likely higher, as the most severely affected dogs may be under-represented in RFGS data. Nevertheless, our estimates derive from the largest datasets currently available for the UK Bulldog, French Bulldog and Pug populations.

Earlier reference values for BOAS prevalence (51% − 64%) were based on cohorts of approximately 200 dogs per breed [17]. In contrast, our study utilised samples that were 5–10 times larger and with notable differences in composition. For example, we observed substantially fewer obese dogs, with the highest percentage in Pug at 17% compared to 61% reported by [17], despite similar median BCS across the two studies. This may indicate that educational activities associated with RFGS testing have influenced management practices related to body condition within participating breed communities, potentially contributing to the lower observed prevalence of BOAS. Our data may also reflect an early response to phenotypic selection, as suggested by lower RFGS grades among breeding dogs in Bulldogs and French Bulldogs.

Historically, RFGS breeding guidelines recommended by the RKC [56] permitted the use of grade 2 dogs for breeding, partly to avoid excessive restriction of the gene pool given high reported BOAS prevalence. However, with the present study indicating that over 75% of tested dogs in each breed are classified as BOAS-, there is scope to further revise these recommendations. Excluding grade 2 dogs from breeding entirely would increase the selection differential, and consequently, could accelerate genetic improvement in respiratory health.

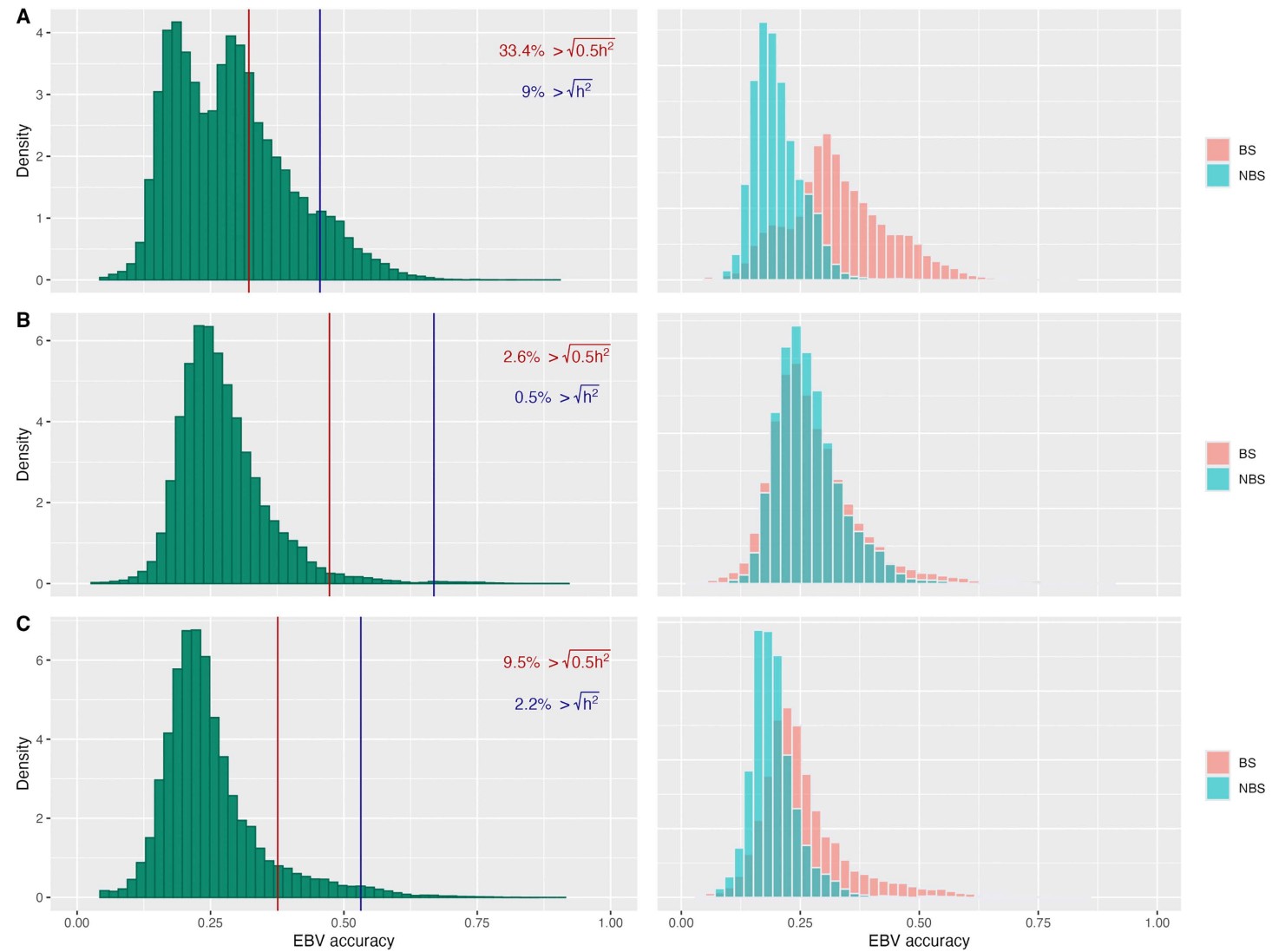

**Fig 5. Distribution of EBV accuracies in Bulldog (A), French Bulldog (B) and Pug (C) born since 2016.** First column shows the distribution of accuracies across all individuals, with vertical lines indicating accuracies expected given selection based on individual's own phenotype ($\sqrt{h^2}$, marked in blue), and accuracy expected given parental average only ($\sqrt{0.5h^2}$, marked in red); with percentages indicating population exceeding given accuracy threshold. Second column illustrates the distribution of EBV accuracies split between BS and NBS dogs.

The presence of significant additive genetic variation in the RFGS grades across all three breeds indicates that within-breed selection is feasible. This study provides the first pedigree-based, within-breed heritability estimates for BOAS. Our estimates are broadly comparable to the across-breed estimate of 0.4 (SE 0.16) reported in Finnish brachyce-phalic breeds based on parent-offspring regression [47]. However, the Finnish populations are smaller and characterised by substantially higher testing uptake. In the present dataset, under-representation of severely affected dogs could restrict phenotypic variance and result in conservative estimates of additive genetic variance. Given differences in population structure, participation rates, methodology, and the relatively large standard error of the Finnish estimate, direct compari-son between studies should therefore be interpreted with caution.

**Table 6. Results of multi-variate analyses.**

| | Bulldog | | | French Bulldog | | | | Pug | | |
|---|---|---|---|---|---|---|---|---|---|---|
| | RFGS grade | BW | STEN grade | RFGS grade | BW | STEN grade | BCS | RFGS grade | BW | STEN grade |
| RFGS grade | 0.17 (0.07) | −0.42 (0.17)** | 0.41 (0.20)* | 0.43 (0.06) | −0.09 (0.08) | 0.59 (0.08)*** | −0.31 (0.14)* | 0.39 (0.09) | 0.14 (0.15) | 0.09 (0.21) |
| BW | −0.06 (0.04) | 0.60 (0.07) | −0.09 (0.14) | 0.00 (0.03) | 0.73 (0.04) | −0.21 (0.09)* | 0.16 (0.11) | 0.02 (0.07) | 0.52 (0.12) | 0.03 (0.17) |
| STEN grade | 0.20 (0.03)*** | −0.04 (0.04) | 0.31 (0.07) | 0.36 (0.03)*** | −0.08 (0.03)** | 0.39 (0.07) | −0.23 (0.14) | 0.13 (0.05)*** | −0.02 (0.05) | 0.32 (0.09) |
| BCS | | | | 0.06 (0.03)** | 0.29 (0.03)*** | 0.02 (0.03) | 0.25 (0.05) | | | |

Diagonal cells marked in grey show heritability; cells above the diagonal show genetic correlations; cells below show phenotypic correlations. All models included sex, linear age, age splines, and coefficient of inbreeding. Colour group was fitted in Bulldog and Pug only. BCS was included as a covariate for grade and BW in Bulldog and Pug, and as a separate trait in French Bulldog. Obesity was fitted for all traits except grade in Pug. Year of test and date of birth were fitted to BW and STEN grade in Bulldog and French Bulldog, and date of birth was also fitted to STEN grade in Pug. Protocol was fitted only for STEN grade in Bulldog, and year of birth only for BW in Bulldog.

Significance is indicated by p < 0.05 (*), p < 0.01 (**), p < 0.001 (***).

The mixed linear model applied here enables more comprehensive partitioning of phenotypic variance than parent-offspring regression by incorporating additional variance components, e.g., permanent environment effect. Both pedigree-based approaches may be influenced by pedigree errors. Genomic methods, which estimate realised genetic sharing, avoid this limitation and improve the response to selection when combined with large-scale phenotyping. To date, BOAS heritability has been estimated using genomic data only once, in a UK sample of approximately 200 dogs per breed [48], yielding an estimate of 0.6. In general, SNP array-based estimates are expected to be lower than pedigree-based estimates because they capture only the proportion of the additive variance tagged by genotyped markers [57]. The lower estimates observed in the present study may reflect the conservative bias related to sampling structure, however, limited methodological detail from [48] prevents firm conclusions regarding the discrepancy.

Significant genetic variance implies that EBV-based selection could be implemented to support more accurate identification of genetic predisposition by incorporating extended pedigree information, and controlling for environmental effects such as the highly significant assessor effect observed here and elsewhere [58].We found that in Bulldog and Pug, the number of dogs with highly accurate EBVs already exceeded the number of phenotyped dogs, illustrating effective dissemination of information from phenotyped dogs through related individuals.

The greatest limitation to both phenotype-based and EBV-based selection remains the low participation in RFGS. Although the percentage of litters from tested parent/s is increasing, the overall engagement remains limited. Danish data indicate promising interest among puppy buyers in purchasing puppies from tested parents [25], however, the same owners remain reluctant to test their own dogs. In the UK, where breed communities have shown limited interest in outcrossing and rely heavily on the within-breed selection approach, increasing participation will be essential for achieving population-wide improvement.

In Denmark, all breeding dogs in Bulldog, French Bulldog and Pug, regardless of pedigree, are legally required to undergo RFGS assessment, with only grades 0 and 1 permitted for breeding [25]. Mandatory health testing can be a powerful tool when supported through national legislation. However, when introduced solely by kennel clubs, such measures risk reducing registration rates rather than increasing participation in health schemes if owners move to unregistered breeding. In the UK, only an estimated 29% of dogs are registered with the RKC [59], and the popularity of RKC-registered dogs has declined in recent years [60]. Maintaining strong connections with the RKC is important, because the RKC-registered communities remain the most directly accessible to health-improvement campaigns.

Our demographic analyses show that the awareness and popularity of health testing is substantially higher among communities involved in conformation showing, demonstrating that those most closely linked to the RKC are also the

groups most consistently exposed to, and engaged with, educational efforts. Nevertheless, even among dogs with all recent ancestors successful in conformation shows, only between 10–20% have been RFGS tested. Participation is expected to increase as the RKC implements new requirements: from 2025, all dogs in Bulldog, French Bulldog and Pug entered at Crufts – the RKC's flagship event and one of the largest dog shows in the world – must undergo RFGS testing, with grade 3 dogs excluded from the competition. From 2026, dogs graded 2 or 3 will no longer be eligible to attend [61].

Conversely, dogs distant from conformation showing and those not fitting breed standard by their coat colour, are under-represented in the tested population, and show slightly worse RFGS grades. There is no evidence that any of the NBS colours are genetically correlated to BOAS, and this is unlikely given the same colour genotype can be considered BS in one breed and NBS in another. For example, black coat colour, determined by CBD103 and TYRP1 [62,63] and common across various dog breeds, is a standard colour in the Pug but non-standard in Bulldogs. Poorer RFGS grades among NBS dogs therefore likely reflect sampling bias, or alternatively, lower levels of health-based selection within NBS breeding communities. Our current sample sizes for NBS dogs are limited, preventing firm conclusions.

Our demographic findings provide additional insight into the proposed strategy of addressing the "brachycephalic crisis" through revisions to breed standards. While breed standards have historically guided conformation goals, adherence is formally assessed only within the conformation show ring. Participation in conformation showing and the use of show dogs in breeding have declined markedly in recent years [60]. Our data shows, for the first time, that the surge in popularity of extreme brachycephalic breeds has been driven largely by dogs that deviate from breed standards at the most obviously visible level of coat colour.

Data from RFGS further indicate the limited influence of breed standard wording on the current UK population. For example, the French Bulldog standard explicitly calls for "well-opened nostrils" [64], a requirement that has remained unchanged in the UK standard since the early 1990's (A. Skipper, personal communication). And yet, our data demonstrate that nostril stenosis remains widespread with only 6% of tested French Bulldog graded as having open nostrils. Thus, while it remains essential that breed standards avoid endorsing exaggerated conformation, changes to wording alone are unlikely to meaningfully influence the wider population, which is already dominated by dogs outside the standard.

The primary efforts should focus on increasing preference for functional traits. As we demonstrate here, selection of breeding stock based on functional traits is feasible. We present, for the first time, estimates of the heritability of nostril stenosis in three brachycephalic breeds, and quantify their genetic correlation with RFGS grade. Although stenotic nares are widely documented as one of the primary conformational contributors to BOAS [17,33,65]), the genetic relationship between those two traits is complex. Notably, in our data the breed with the highest prevalence of stenotic nostrils (French Bulldog) also showed the lowest prevalence of BOAS. Our analyses show that RFGS grade and stenosis are positively genetically correlated, with moderately high estimates, but represent genetically distinct traits. This suggests that selection strategies targeting both traits simultaneously could be more effective than relying on either trait alone.

We also provide the first heritability estimates for body weight in the three extreme brachycephalic breeds. Although body weight is known to be highly heritable in humans, estimates for dogs remain scarce and have not previously included these breeds [66,67]. Our moderately high estimates fall near the upper range of those reported for non-brachycephalic breeds in Sweden [65], and are substantially higher than the heritability of 0.18 reported for adult weight in Boxers [66]. As the body weight observations in our study were corrected for obesity, and for BCS in Bulldog and Pug, these estimates likely represent the genetic determinism of body size rather than fat deposition.

Although BCS is often considered a primarily management-related trait influenced by factors such as diet [68], exercise [69], or neutering [70], genetic contributions to this trait have been reported. In Labrador Retrievers, BCS showed a heritability of 0.7 and an association with the *DENND1B* gene [71], though this effect appears breed-specific. In our dataset, additive genetic variance for BCS was detected only in French Bulldog, despite clear phenotypic variation in all

three breeds. This may reflect limited sample size, particularly for Pug, or potentially a lower genetic determinism of BCS in these breeds.

Obesity and high BCSs are frequently cited as risk factors for BOAS [9,10,16]. Using a multivariate animal model, we explored the genetic relationships between body size measures and RFGS grade, and detected several statistically significant genetic correlations. Counterintuitively, the significant estimates between RFGS grade and BW in Bulldog, and between RFGS grade and BCS in French Bulldog, were negative. Because both BW and BCS had been corrected for obesity, these results suggest that larger, but not obese, dogs may be less genetically predisposed to BOAS. However, it is important to note that dogs severely affected with BOAS often experience regurgitation [14], which can reduce BCS and potentially bias phenotypic observations. In a previous analysis of French Bulldog, nearly 10% of RFGS grade 3 dogs were scored as underweight, which was attributed to regurgitation [17] (Liu et al., 2017). Additional data, especially in Pug where none of the genetic correlations reached statistical significance, are needed to clarify the pleiotropic relationships between body size and respiratory function. Taken together with the significant genetic correlations among body weight, BCS, RFGS grade and nostril stenosis, our findings suggest that selection against BOAS could be more effective when all four traits are recorded and evaluated jointly within a multivariate framework.

Our findings have further implications for the organisation of the RFGS scheme. Although age at testing had a significant effect on the RFGS grade in our data, the magnitude of this effect was extremely small, consistent with previous studies showing minimal or no influence of age [10,11,15,16]. Combined with high repeatability estimate for Bulldog and French Bulldog, these results raise questions about the necessity of repeated measurements throughout a dog's life. While veterinary care for BOAS is typically sought between 3 (Monnet, 2011) and 6 [72] years of age, this reflects owner decision-making rather than appearance of clinical signs detectable by trained RFGS assessors.

Although our findings substantially advance the understanding of BOAS and provide refined guidance for developing effective selection strategies, several limitations should be considered. First, the data used here were obtained through voluntary participation, which may introduce sampling bias and limit the representativeness of the wider UK population. Second, while BOAS remains the most recognised and extensively studied component of the brachycephalic syndrome, it does not encompass its full clinical spectrum [73]. Other conditions, including ocular disease [74], middle ear effusions [75] or gastrointestinal abnormalities [76] also contribute to the overall welfare burden. Some, particularly sleep and gastrointestinal issues, may improve alongside better respiratory function, but comprehensive evaluation of the entire brachycephalic syndrome was beyond the scope of this study.

## Conclusions

This study analysed the largest RFGS dataset to date and shows that BOAS prevalence in RKC registered UK Bulldogs, French Bulldogs and Pugs is lower than previously reported, supporting refinement of current RFGS breeding recommendations. We identified substantial genetic variance in RFGS grade, stenotic nares and body weight, as well as meaningful genetic correlations among these traits, demonstrating that within-breed selection may improve respiratory health, particularly when multiple BOAS-related traits are recorded and evaluated together. However, the impact of any selection programme will depend on the levels of participation. Engagement with RFGS testing remains low, especially outside show-associated subpopulations, and increasing uptake is essential to achieve population-wide improvements in respiratory function.

## Supporting information

**S1 File. Details of the respiratory function grading scheme.**
(DOCX)

**S2 File. Preliminary estimation of heritability of RFGS Grade using parent offspring regression.**
(DOCX)

**S1 Fig. PCA plot of Bulldog population, stratified by year of birth and coat colour group.**
(GIF)

**S2 Fig. PCA plot of French Bulldog population, stratified by year of birth and coat colour group.**
(GIF)

**S3 Fig. PCA plot of Pug population, stratified by year of birth and coat colour group.**
(GIF)

**S1 Table. Coat colours in the three breeds divided into breed standard, and non-breed standard.**
(XLSX)

**S2 Table. Residual and assessor effect correlation between traits used in multivariate analyses.**
(DOCX)

**S3 Table. Pedigree for Bulldog.**
(CSV)

**S4 Table. Pedigree for French Bulldog.**
(CSV)

**S5 Table. Pedigree for Pug.**
(CSV)

**S6 Table. Phenotypes for Bulldog.**
(CSV)

**S7 Table. Phenotypes for French Bulldog.**
(CSV)

**S8 Table. Phenotypes for Pug.**
(CSV)

## Acknowledgments

The authors would like to thank colleagues at the RKC for their ongoing support, as well as day to day management of the RFGS. We are also very grateful to Nai-Chieh Liu DVM MPhil PhD and Fran Tomlinson MA VetMB MRCVS for their dedication and expertise in the development of the RFGS. Finally, we thank the breeders and owners who presented their dogs for testing. Their participation is essential in generating the data needed to improve opportunities for selection against the disease and, ultimately, to support better health outcomes in their breeds.

## Author contributions

**Conceptualization:** Joanna Jadwiga Ilska, Fern McDonnell.

**Data curation:** Joanna Jadwiga Ilska, Fern McDonnell.

**Formal analysis:** Joanna Jadwiga Ilska.

**Investigation:** Joanna Jadwiga Ilska.

**Methodology:** Joanna Jadwiga Ilska.

**Project administration:** Fern McDonnell.

**Software:** Joanna Jadwiga Ilska.

**Visualization:** Joanna Jadwiga Ilska.

**Writing – original draft:** Joanna Jadwiga Ilska.

**Writing – review & editing:** Joanna Jadwiga Ilska, Fern McDonnell, Jane Frances Ladlow.

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
