## [Decision Letter · Decision Letter 0]

12 Feb 2026

Dear Dr. Ilska,

We look forward to receiving your revised manuscript.

Kind regards,

Adolfo Maria Tambella, DVM, MSc

Academic Editor

PLOS One

3. In the online submission form, you indicated that [The pedigree records and RFGS results of individual dogs registered with the Royal Kennel Club, are available on reasonable request in anonymised format].

Reviewers' comments:

Reviewer's Responses to Questions

**Comments to the Author**

1. Is the manuscript technically sound, and do the data support the conclusions?

Reviewer #1: Yes

Reviewer #2: Yes

Reviewer #3: Yes

2. Has the statistical analysis been performed appropriately and rigorously?

Reviewer #1: Yes

Reviewer #2: Yes

Reviewer #3: Yes

3. Have the authors made all data underlying the findings in their manuscript fully available?

Reviewer #1: No

Reviewer #2: No

Reviewer #3: Yes

4. Is the manuscript presented in an intelligible fashion and written in standard English?

Reviewer #1: Yes

Reviewer #2: Yes

Reviewer #3: Yes

Reviewer #1: Ilska et al. have studied three dog breeds susceptible to Brachycephalic Obstructive Airway Syndrome (BOAS). Using pedigrees and phenotype data, they have characterised these dog breeds and have estimated the breed-specific heritabilites for BOAS. This is a well-written original study, providing evidence on which future policies or breeding decisions may be based. I think this manuscript is suitable for publication. I have only a few comments.

General comments

Levels of genetic differentiation from PCA plots: I am delighted that Ilska and colleagues used our package for pedigree PCA. I would like to point out that ‘fanning out’ over time is a common observation when the founders of a pedigree are assumed to be unrelated. See fig. 1 (bottom-left panel) of our paper (Lee et al. 2025). On its own, this is not evidence for an increasing genetic differentiation. But it also does not preclude the possibility of increasing differentiation. To assess genetic differentiation from a pedigree, it might be more useful to compare the distributions of expected relationships between the coat colour groups from the pedigree A matrix. This could be done in time slices.

The main outcome of this work is the heritably estimated for BOAS. This is good and important. But I think it would be important to discuss potential biases to these estimates. E.g. Do litter mates share a common environment which may cause upward bias to h^2 estimates? Also, BOAS is linked to BCS, and BCS may be similar between relatives if these share the same environment?

At the same time there may be too low reporting of BOAS, or perhaps BOAS may only be detected of a dog is presented for other reasons. May this cause a downward bias to h^2 estimates.

Specific

• P10L225 data need not be normally distributed. The residuals in least-squares regression analyses should be. The effect is the same though. Much of the data are not continuous/they are ordinal. Nonparametric approaches as they were mainly used are other the better choice here.

• L235 number of litters is discrete, two-sample Wilcoxon test might perhaps be more appropriate than t-test here?

• L452 Perhaps better ‘Heredity estimation’ or similar? Better to describe section content rather than the method used

• L492 Perhaps better ‘Genetic correlations’ or similar?

Reviewer #2: Comments to Authors: Ilska, et al., provide a very important paper analyzing the largest (to date) dataset of RFGS data. The Respiratory Function Grading Scheme was established to objectively characterize individual dogs’ level of clinical signs associated with Brachycephalic Obstructive Airway Syndrome, which is of high concern due to its welfare impacts on dogs. The paper is timely and very well-written. They have calculated heritability estimates for RFGS grade and nostril stenosis, and with their data have been able to draw numerous interesting conclusions, some of which are a bit tangential but still fascinating, including the fact that so many dogs in the UK do not conform to breed standards, that participation in health testing is higher in dogs linked to confirmation-show lines (which means that “pet” breeders are not participating as much!), and others. Their data helps clarify the scope of breeding recommendations within the RFGS scale and even points to modifying the selection strategies (targeting multiple of these phenotypes at once can create more simultaneous progress). The paper was delightful to read and only needs a few relatively minor changes.

• There is some concern that the authors indicate that they will not make their data fully available; this does not seem to comply with journal policy. If the pedigrees are only provided in an anonymized format, will they even be useful to other researchers? I’m not certain if this would fall into the PLOS “rare exceptions” category under their Data Availability policy.

o I also wonder if the “custom Fortran 90 program” (line 210) is shareable?

• Lines 143-147: are the “two currently available estimates of heritability” for BOAS (as a phenotype)? Or for some other phenotype?

o The data from these two mentioned studies should be included as an additional paragraph in your discussion, comparing/contrasting with your own present findings. Furthermore, it is important to at least mention the fact that your heritability calculations are pedigree based, whereas the Sargan 2021 group used genetic data (although only from 200 dogs). SNP-based heritability is typically expected to yield lower heritabilities, due to missing rare variants, and pedigree-based methods (relying on assumed genetic sharing, e.g., 50% for full siblings) ignores that actual sharing can vary due to random segregation. The latter (pedigree-based heritabilities) are also subject to pedigree errors, which we all know happen more frequently than anyone would like to admit in dogs. Therefore, an entire paragraph should be added to the discussion explaining these pros/cons, and comparing the strengths/weaknesses of the present data to those other previously published studies. Given all of the above, it would also be interesting to have the authors speculate on why their heritability estimates are actually LOWER than the one existing SNP-based heritability estimate? (It could be that this cannot be done, as I’m not sure the Sargan 2021 publication had all the needed details?). Finally, the paragraph can wrap with a suggestion that, ideally, future work would have thousands of phenotyped samples, as in your study, that ALSO had genome-wide data, where both pedigree and actual genetic data could be combined, for a hopefully even more accurate estimate.

• Figure 3 (the histogram of body weight and body condition score) is not actually referred to anywhere in the text? And line 428, which calls out Figure 3…I think this should say Figure 4?

• Not an issue, but I just want to say that the PCA plots by birth year (Supps 4-6) are SO COOL!

• Table 5: What does the bolding indicate on this table? I think it’s to do with the p-values being less than 0.05, but this should be indicated in the table legend.

• On both Table 6 and Table S6, can you please include in the table legend what the greyed boxes indicate? I think those are your “diagonal cells”, but this should just be made clear.

• Line 675: it would probably be best to add “nostril” before the word stenosis here. I know you’re talking about nostril stenosis throughout the manuscript…but since other things can be stenotic, it’s better to just be abundantly clear.

• Line 686: Please change “symptoms” to “clinical signs”. Since symptoms are something that needs to be communicated with words (e.g., my head hurts, I feel very nauseated), none of our veterinary patients (or human pediatric patients, for that matter) can ever have symptoms. We can only measure and observe clinical signs in patients that cannot speak.

Reviewer #3: Thank you for an interesting and large heritability analysis of BOAS traits. This manuscript is well written and well thought through. I don't have much to recommend! It was readable, pertinent to our patients, and interesting.

The only big comment I have about the manuscript is the word "genetic" in the title. While this is true because we're talking about heritability, I think that many people will assume genomic sequencing rather than evaluation through pedigree- I did! I think that this one change (in the title) will set reader expectations.

In Table 6- please state what the shading indicates.

thanks for an interesting study!

.

Reviewer #1: No

Reviewer #2: No

Reviewer #3: No

---

## [Author Response · Author response to Decision Letter 1]

16 Mar 2026

We thank the Reviewers for the positive reception of the manuscript and for the constructive feedback. Our point-by-point responses to their comments are outlined below. The line numbers noted below refer to the “paper post review with tracked changes.docx”.

Reviewer 1

"Ilska et al. have studied three dog breeds susceptible to Brachycephalic Obstructive Airway Syndrome (BOAS). Using pedigrees and phenotype data, they have characterised these dog breeds and have estimated the breed-specific heritabilites for BOAS. This is a well-written original study, providing evidence on which future policies or breeding decisions may be based. I think this manuscript is suitable for publication. I have only a few comments.

General comments:

Levels of genetic differentiation from PCA plots: I am delighted that Ilska and colleagues used our package for pedigree PCA. I would like to point out that ‘fanning out’ over time is a common observation when the founders of a pedigree are assumed to be unrelated. See fig. 1 (bottom-left panel) of our paper (Lee et al. 2025). On its own, this is not evidence for an increasing genetic differentiation. But it also does not preclude the possibility of increasing differentiation. To assess genetic differentiation from a pedigree, it might be more useful to compare the distributions of expected relationships between the coat colour groups from the pedigree A matrix. This could be done in time slices."

We thank the reviewer for underlining the possibility of the fanning out being due to founders being included in pedigree. We have now included this caveat in the manuscript in lines 443 - 445. However, the separation observed in our data was not solely temporal. This is supported by dogs from different coat colour groups clustering together, as well as strong association between testing participants and show breeding, which support the hypothesis of emerging substructure, rather than a purely generational artefact.

"The main outcome of this work is the heritably estimated for BOAS. This is good and important. But I think it would be important to discuss potential biases to these estimates. E.g. Do litter mates share a common environment which may cause upward bias to h^2 estimates? Also, BOAS is linked to BCS, and BCS may be similar between relatives if these share the same environment? At the same time there may be too low reporting of BOAS, or perhaps BOAS may only be detected of a dog is presented for other reasons. May this cause a downward bias to h^2 estimates. "

Following the comments of Reviewers 1 and 2, we have expanded on the impact that under-representation of affected individuals may have on the estimates of heritability (lines 600-609).

At present, our dataset contains few phenotyped littermates, which precluded inclusion of a litter effect in the model. We intend to revisit this as additional data become available, allowing more comprehensive modelling of common environmental influences, including litter and potential maternal effects. However, we do not anticipate that such effects would substantially influence the traits analysed here. RFGS is assessed in adult dogs, well after weaning, and in contrast to livestock, related dogs rarely share a long-term environment, as puppies are typically rehomed at 8–12 weeks of age. Consequently, persistent shared environmental effects among relatives are expected to be limited.

Specific

• P10L225 data need not be normally distributed. The residuals in least-squares regression analyses should be. The effect is the same though. Much of the data are not continuous/they are ordinal. Nonparametric approaches as they were mainly used are other the better choice here.

We have amended the wording in lines 226-227, describing the traits as ordinal.

• L235 number of litters is discrete, two-sample Wilcoxon test might perhaps be more appropriate than t-test here?

We have opted for t-test with unequal variances (Welch test) due to unequal variances and unequal sample sizes, which were both present in our data, and which can lead to increase in Type I error rate under Wilcoxon test.

• L452 Perhaps better ‘Heredity estimation’ or similar? Better to describe section content rather than the method used

We have changed the heading to Genetic parameters of RFGS grade (line 469)

• L492 Perhaps better ‘Genetic correlations’ or similar?

We have changed the heading to Correlations Between RFGS grade and Conformation Traits (line 515).

Reviewer 2:

"Comments to Authors: Ilska, et al., provide a very important paper analyzing the largest (to date) dataset of RFGS data. The Respiratory Function Grading Scheme was established to objectively characterize individual dogs’ level of clinical signs associated with Brachycephalic Obstructive Airway Syndrome, which is of high concern due to its welfare impacts on dogs. The paper is timely and very well-written. They have calculated heritability estimates for RFGS grade and nostril stenosis, and with their data have been able to draw numerous interesting conclusions, some of which are a bit tangential but still fascinating, including the fact that so many dogs in the UK do not conform to breed standards, that participation in health testing is higher in dogs linked to confirmation-show lines (which means that “pet” breeders are not participating as much!), and others. Their data helps clarify the scope of breeding recommendations within the RFGS scale and even points to modifying the selection strategies (targeting multiple of these phenotypes at once can create more simultaneous progress). The paper was delightful to read and only needs a few relatively minor changes.

• There is some concern that the authors indicate that they will not make their data fully available; this does not seem to comply with journal policy. If the pedigrees are only provided in an anonymized format, will they even be useful to other researchers? I’m not certain if this would fall into the PLOS “rare exceptions” category under their Data Availability policy."

We provide the data where the phenotypes and associated variables can be matched to anonymised pedigree in the supplementary materials. This would enable researchers to repeat our analysis, and/or apply different models to the data.

• I also wonder if the “custom Fortran 90 program” (line 210) is shareable?

The analyses were conducted using a custom Fortran program developed for internal use, optimised for the data structures used in RKC database. The code is not currently available for distribution, but its functionality can be found in other, publicly available software packages.

• Lines 143-147: are the “two currently available estimates of heritability” for BOAS (as a phenotype)? Or for some other phenotype?

We have added that it is heritability of BOAS.

• The data from these two mentioned studies should be included as an additional paragraph in your discussion, comparing/contrasting with your own present findings. Furthermore, it is important to at least mention the fact that your heritability calculations are pedigree based, whereas the Sargan 2021 group used genetic data (although only from 200 dogs). SNP-based heritability is typically expected to yield lower heritabilities, due to missing rare variants, and pedigree- based methods (relying on assumed genetic sharing, e.g., 50% for full siblings) ignores that actual sharing can vary due to random segregation. The latter (pedigree-based heritabilities) are also subject to pedigree errors, which we all know happen more frequently than anyone would like to admit in dogs. Therefore, an entire paragraph should be added to the discussion explaining these pros/cons, and comparing the strengths/weaknesses of the present data to those other previously published studies. Given all of the above, it would also be interesting to have the authors speculate on why their heritability estimates are actually LOWER than the one existing SNP-based heritability estimate? (It could be that this cannot be done, as I’m not sure the Sargan 2021 publication had all the needed details?). Finally, the paragraph can wrap with a suggestion that, ideally, future work would have thousands of phenotyped samples, as in your study, that ALSO had genome-wide data, where both pedigree and actual genetic data could be combined, for a hopefully even more accurate estimate.

We have now expanded on our comparison to the Finnish study (lines 596 – 611), and added a paragraph comparing our estimates to those of Sargan (2021) (lines 617 - 624). Unfortunately, lack of details in Sargan (2021) makes drawing any firm conclusions impossible.

• Figure 3 (the histogram of body weight and body condition score) is not actually referred to anywhere in the text? And line 428, which calls out Figure 3…I think this should say Figure 4?

Thank you for spotting this. Figure 3 is referred to in lines 408 and 410, previously erroneously citing Figure 4. Line 438 was corrected to Figure 4.

• Not an issue, but I just want to say that the PCA plots by birth year (Supps 4-6) are SO COOL!

Thank you – we were very excited to see those too!

• Table 5: What does the bolding indicate on this table? I think it’s to do with the p-values being less than 0.05, but this should be indicated in the table legend.

Reviewer is correct, bolding indicates statistically significant effects. We have added the explanation in the table legend.

• On both Table 6 and Table S6, can you please include in the table legend what the greyed boxes indicate? I think those are your “diagonal cells”, but this should just be made clear.

We have corrected the legends as suggested.

• Line 675: it would probably be best to add “nostril” before the word stenosis here. I know you’re talking about nostril stenosis throughout the manuscript…but since other things can be stenotic, it’s better to just be abundantly clear.

We have added “nostril” as suggested (line 742).

• Line 686: Please change “symptoms” to “clinical signs”. Since symptoms are something that needs to be communicated with words (e.g., my head hurts, I feel very nauseated), none of our veterinary patients (or human pediatric patients, for that matter) can ever have symptoms. We can only measure and observe clinical signs in patients that cannot speak.

We have amended this as suggested (line 753).

Reviewer #3:

Thank you for an interesting and large heritability analysis of BOAS traits. This manuscript is well written and well thought through. I don't have much to recommend! It was readable, pertinent to our patients, and interesting. The only big comment I have about the manuscript is the word "genetic" in the title. While this is true because we're

talking about heritability, I think that many people will assume genomic sequencing rather than evaluation through pedigree- I did! I think that this one change (in the title) will set reader expectations.

We have amended the title from “Genetic variation” to “Quantitative genetic analysis”, as we agree with Reviewer’s comment on setting appropriate expectation for the reader.

In Table 6- please state what the shading indicates.

Thank you – we have amended the legends for Table 6 and S6.

thanks for an interesting study!

---

## [Decision Letter · Decision Letter 1]

10 Apr 2026

Quantitative genetic analysis of respiratory function and related traits in Bulldogs, French Bulldogs and Pugs

PONE-D-25-67266R1

Dear Dr. Ilska,

We’re pleased to inform you that your manuscript has been judged scientifically suitable for publication and will be formally accepted for publication once it meets all outstanding technical requirements.

Kind regards,

Adolfo Maria Tambella, DVM, MSc

Academic Editor

PLOS One

Additional Editor Comments (optional):

The manuscript has been entirely and carefully revised. The current version can be considered suitable for publication in PLoS ONE. Congratulations to the authors!

Reviewers' comments:

Reviewer's Responses to Questions

**Comments to the Author**

Reviewer #1: All comments have been addressed

Reviewer #3: All comments have been addressed

2. Is the manuscript technically sound, and do the data support the conclusions?

Reviewer #1: Yes

Reviewer #3: Yes

3. Has the statistical analysis been performed appropriately and rigorously?

Reviewer #1: Yes

Reviewer #3: Yes

4. Have the authors made all data underlying the findings in their manuscript fully available?

Reviewer #1: No

Reviewer #3: Yes

5. Is the manuscript presented in an intelligible fashion and written in standard English?

Reviewer #1: Yes

Reviewer #3: Yes

Reviewer #1: (No Response)

Reviewer #3: Thanks for the changes. It looks good. Was the legend to table 6 updated - maybe I missed this since you said you did it in the response? Please include what the grey cells mean. In table 5, please include what the bold means.

.

Reviewer #1: No

Reviewer #3: No

---

## [Editor Report · Acceptance letter]

PONE-D-25-67266R1

PLOS One

Dear Dr. Ilska,

I'm pleased to inform you that your manuscript has been deemed suitable for publication in PLOS One. Congratulations! Your manuscript is now being handed over to our production team.

Kind regards,

on behalf of

Prof. Adolfo Maria Tambella

Academic Editor

PLOS One